

# The Effects of Younger Dryas Orbital Parameter and Atmospheric pCO$_2$ Changes on Radiative Forcing and African Monsoonal Circulation

Taylor M. Hughlett[1], Arne M.E. Winguth[1], Nan Rosenbloom[2]

[1]Earth and Environmental Sciences, University of Texas Arlington, Arlington, 76013, United States of America
[2]National Center for Atmospheric Research, Boulder, 80305, United States of America

*Correspondence to*: Taylor M. Hughlett (taylor.hughlett@uta.edu)

**Abstract.** Differences in the Atlantic meridional overturning circulation (AMOC) from the Younger Dryas (YD) to the Holocene can be explained by, but not limited to, factors relating to atmospheric greenhouse gas concentrations, discharge of freshwater into the surface ocean, and changes in Earth's orbital parameters. Utilizing the Community Earth System Model (CESM1.0.5) with moderate resolution, this study evaluates how Younger Dryas seasonal and annual radiative forcing affect the climate change and variability. The Younger Dryas to Holocene changes in radiative forcing are mostly attributed to change in orbital parameters and to lesser extent to the relatively small rise in atmosphere pCO$_2$, which is supported by a comparison of model simulations with proxy reconstructions of sea surface temperature and oceanic δ$^{18}$O. These factors led to increased precipitation and reduced transport of water masses in the North Atlantic Ocean. Atmospheric pCO$_2$ and orbital parameter changes are not substantial enough to explain the transition to the Younger Dryas northern hemispheric cooling. Younger Dryas to Holocene changes in the Monsoonal circulation over the African continent appears to be more affected by changes in orbital parameters than in atmospheric pCO$_2$ but underestimated compared to observed reconstructions from ice and sediment cores. Thus, additional mechanisms such as fresh water hosed-cooling and/or ice sheet-albedo effect need to be considered to explain the Younger Dryas to Holocene climate change and variability.

## 1 Introduction

The Younger Dryas (YD) cooling event (~12.9 kya) is one of the most recent periods of abrupt climate change in the geological record. It is characterized by a transition to near-glacial conditions for the Northern Hemisphere as well as a decrease in snow accumulation, dust accumulations, and the Atlantic meridional overturning circulation (AMOC; Alley, 2000; Severinghaus et al., 1998). The most widely accepted hypothesis for this cooling event is a reduction of the AMOC due to a pulse of freshwater from Lake Agassiz at the end of the Bølling-Allerød warm period (Broecker and Denton, 1989; Rooth, 1982). A freshwater pulse into the Northern Atlantic at the area of deep-water formation causes a density imbalance that would lead to stratification of the water masses and a decrease in the strength of the overturning circulation (Rooth, 1982), poleward heat transport, and



North Atlantic cooling (e.g. Seager and Battisti, 2007; Peltier, 2007; Cheng et al., 2011). Freshwater was potentially discharged to the Northern Atlantic Ocean by three possible routes include by a switch from the southern route through the Mississippi River during Bølling-Allerød to an eastern route (Broecker et al., 1989), or to a northern route through the Mackenzie River (Tarasov and Peltier, 2005; Murton et al., 2010). Recent sedimentary analysis favours the northern route as a probable route

for freshwater discharge into the Northern Atlantic Ocean (Murton et al., 2010). Little geologic evidence has been established for freshwater forcing routes through the eastern route (Broecker, 2006; Teller et al., 2005).

Two major radiative forcings, changes in the Earth orbital parameters and greenhouse gas concentration affect the Younger-Dryas to Holocene in the climate (Kutzbach et al., 1998) that will be discussed in following. Other changes in radiative forcing

such as aerosols forcing will be not be the focus of this study and we refer to study of Muchitiello et al., (2017). Variations in orbital parameters of the Earth's path around the sun have been also considered as factors leading to abrupt climate (e.g. Broecker 1989, Peltier et al., 2006). Changes in the Younger Dryas external climate forcings are affected, among others, by the combination of the three orbital parameters: precession, obliquity, and eccentricity. Precession of the equinoxes, defined as the changing position of the equinoxes on the elliptic pathway of the Earth around the sun, encompasses a 23,000-19,000-

year cycle (Berger, 1988; Crowley and North, 1991). Obliquity describes the degree of Earth's axial tilt toward the sun and has a 41,000 and 54,000 -year cycle (Berger, 1988). Eccentricity has a 413,000, 123,000, and 95,000 cycle, and involves the shape of the Earth's elliptical orbit around the sun (Berger, 1988). The precession of the equinox, modulated by eccentricity, influences the strength of the monsoonal circulation (Prell and Kutzbach, 1978). The precessional maxima occurs when summer solar radiation increases over the subtropics, leading to an amplification of the  warming and a subsequent decrease

of the low-pressure area  over land (Kutzbach, 1981, Zhisheng et al., 1993). Over the adjacent ocean a high-pressure area forms and warm, humid marine air is advected toward the land. Increased advection leads to enhanced convection and subsequent monsoonal precipitation increase. During the winter, the incoming solar radiation is weaker during precessional maxima compared to normal conditions; leading to a high-pressure area over the land due to cooling temperatures and a low-pressure area over the warmer ocean. The precession of the equinox and the eccentricity during the Younger Dryas was more extreme

compared to preindustrial times, and thus monsoonal circulation was likely stronger (Table 1; Kutzbach et al., 1998).

The Earth's obliquity contributes to the amplitude of the seasonal cycles, particularly in polar latitudes. For example, when Earth's obliquity increases, seasonality in the polar regions becomes more amplified due to reduced solar radiation during the winter and increase during the summer. The solar insolation during the YD is estimated to be approximately 1 W m$^{-2}$ lower

than the preindustrial value of 1,365 W m$^{-2}$ (Table 1; Otto-Bliesner et al., 2006b). Thus, seasonal climate variations were likely more extreme during the YD compared to the late Holocene, and it has been estimated that the summer (winter) insolation was 28 W m$^{-2}$ (18 W m$^{-2}$) higher (lower) during the YD compared to present day (Kutzbach et al., 1998).





Decrease in obliquity potential influences the strength of the Atlantic meridional overturning circulation; for example, when obliquity decreases from 22.8° to 22.4°, the AMOC transitions from ~26 Sv to ~17 Sv (strong state and intermediate state respectively; Friedrich et al., 2009). In addition, simulations with orbital forcing similar to 6 ka indicate an increase in AMOC strength, a cooling in the high latitudes, and a southward-migration of the ITCZ (Fischer and Jungclaus, 2010).

The seasonal northward and southward migration of the Intertropical convergence zone (ITCZ) regulates the West African monsoonal system (Chiang et al., 2002; Griffiths, 1972). During the winter, North Africa is covered with a high-pressure system while the equatorial region of Africa is covered in a low-pressure system, and the ITCZ is located north of the equator (Griffiths, 1972). The low-pressure system migrates northward in the spring to cover the northern desert, and the ITCZ follows

to bring precipitation to the Sahel, the region just south of the desert (Folland et al., 1986; Gasse, 2000; Griffiths, 1972; Kröpelin et al., 2008; Nicholson, 2013). The ITCZ continues to shift northward through the summer until it reaches its northernmost location at approximately 20°N latitude (Griffiths, 1972).

Lacustrine and terrestrial sediment cores (Garcin et al., 2007; Gasse, 2000; Talbot et al., 2007) represent the West African

monsoon from different regions. Pollen assemblages, from the Tanzanian Lake Masoko sediment core, suggest a northward migration of the ITCZ over the African continent, which resulted in the decline of the southern African monsoon intensity for the region (Garcin et al., 2007). The arid period for the southern region, specifically south of 10 °S, is also represented in lacustrine sediment cores from the area (Gasse, 2000; Talbot et al., 2007). With the northward migration of the ITCZ toward the northern tropics precipitation increased over the region: supported by lacustrine cores from Lakes Tangayika, Bosumtwi,

and Malawi (Talbot et al., 2007), as well as pollen records from the Niger River mouth (Lezine et al., 2005). This increased precipitation is known as the "African Humid Period". During this period, it is likely that increased solar radiation induced from changes in orbital parameters resulted in a strengthening of the Northern African summer monsoon (Kutzbach, 1981; Kutzbach and Guetter, 1986), while the Southern African monsoon weakened.

The other important radiative forcing discussed in this study is the change in greenhouse gas radiative forcing between the Younger Dryas and the preindustrial period. Ice cores from Antarctica suggest large $CO_2$ emissions at the termination of the last three deglaciations could have led to the collapse of the AMOC (Broecker, 1997; Fischer et al., 1999). Climate simulations support this finding that at atmospheric $pCO_2$ concentrations equal to half of present-day values, such as LGM concentrations, would lead to a near shutdown of the AMOC (Wood et al., 1999; Hu et al., 2004; Rahmstorf, 2000; Gregory et al., 2005;

Stouffer et al., 1989; Manabe and Stouffer, 1999; Klockmann et al., 2015; Manabe and Stouffer, 1994; Voss and Mikolajewicz, 2001). Decline in greenhouse gas induced radiative forcing and associated tropospheric cooling could lead to higher aridity by decreased precipitation (due to the temperature dependence of vapor saturation pressure). The $CO_2$ radiative effect (cooling and increase in salinity) influences convection in the GIN Sea and the AMOC strength (Broecker and Denton, 1989; Rooth, 1982).





Decrease of greenhouse gas radiative forcing during the last deglaciation may have altered African monsoonal circulation by precipitation increases in the southern area of the Sahel whereas with orbital forcing changes, precipitation increases over the entire Sahel region and expands northward over the northern area of the African continent (Otto-Bliesner et al., 2014).

In the following, the response AMOC and West African Monsoon circulation and their intensity to changes in the orbital and greenhouse gas radiative forcing during Younger Dryas.

## 2 Methods

### 2.1 Model Description

The simulations for this study utilize the National Center for Atmospheric Research's Community Earth System Model version 1.0.5 (NCAR CESM1.0), a comprehensive climate model of intermediate complexity (Gent et al., 2011). CESM1 contains a central coupler (CPL7) that allows for the exchange of information between the four components, the Community Land Model version 4 (CLM4; Oleson et al., 2010), The Community Atmosphere Model version 4 (CAM4; Neale et al., 2010), the Los Alamos Sea Ice Model (CICE; Hunke et al., 2013), and the Parallel Ocean Program version 2 (POP2; Smith et al., 2010).

Additional information regarding the model and the model components can be obtained from the CESM website (www2.cesm.ucar.edu). The model resolution chosen for this study is the f09_g16, which has a 1° by 1° grid, with 26 and 60 vertical layers in the atmosphere and the ocean respectively.

Boundary and initial conditions for the preindustrial 1850 control simulation and for the Younger Dryas are summarized in

Table 1. Orbital parameters are calculated within CESM1 after Neale et al. (2010) as a function of the calendar year referenced to January 1, 1950 (considered present day); therefore, any date before January 1, 1950 is referenced as before present (BP). The orbital year chosen for the preindustrial simulation was 1990 (Brady et al., 2013). For the Younger Dryas simulations, the orbital year has been set to the equivalent of 13,000 yrs BP. The change in radiative forcing between the 13 ka orbital parameters and the preindustrial orbital parameters is illustrated in figure 1. Greenhouse gas concentrations for the preindustrial

simulation are 284.7 ppmv, 791.6 ppbv, and 275.68 ppbv for $CO_2$, $CH_4$, and $N_2O$ respectively (Table 1; Brady et al., 2013). For the Younger Dryas the greenhouse gas concentrations are simulated at 237.57 ppmv, 632.0 ppbv, and 265.0 ppbv for $CO_2$, $CH_4$, and $N_2O$ respectively (Table 1; Joos and Spahni, 2008).

Atmospheric $pCO_2$ was lower during the Younger Dryas compared to 1850 by approximately 47.13 ppmv (Joos and Spahni,

2008a; Table 3-1), which would result in a global decline in radiative forcing of 1.04 W m$^{-2}$. This decrease would correspond to a 0.76°C cooling (Bitz et al., 2011; Danabasoglu and Gent, 2009; Hansen et al., 1988; IPCC, 2007; Manabe and Bryan Jr, 1985; Winguth et al., 2015), but is relative small compared to the large seasonal changes in orbital forcing. During the summer



insolation increases between the Younger Dryas and the Holocene by approximately 50 W m$^{-2}$. Insolation during the winter decreases by approximately 50 W m$^{-2}$ between the Younger Dryas and the Holocene. Therefore, it is expected that the seasonal variability of the climate would be more extreme for changes in orbital forcing compared to changes in atmospheric $pCO_2$.

The preindustrial 1850 control and the Younger Dryas model boundary and initial conditions are reported in Table 4-1. Incoming solar radiation for the preindustrial simulation was set to 1,366 W m$^{-2}$ and adjusted for the Younger Dryas to 1,365 W m$^{-2}$ based on calculations for the Last Glacial Maximum (Brady et al., 2013; Otto-Bliesner et al., 2006a). Modern ice sheets were used for the preindustrial simulation.

### 2.2 Experiment Design

Three sensitivity experiments were used to determine the sensitivity of the climate to Younger Dryas boundary conditions. A preindustrial control experiment (PI) used the boundary conditions of the year 1850 (Brady et al. (2013) and was initialized from a 1000-year simulation performed by NCAR (Table 2). A 100-year integration period after boundary and initial condition input was used for the PI simulation. The first sensitivity experiment (YDORB) utilized preindustrial boundary and initial conditions for greenhouse gases, ice sheets, and vegetation, while orbital parameters were set to 13,000 years before present

(Table 2). The second sensitivity experiment (YDCO2) used preindustrial boundary and initial conditions for orbital parameters, ice sheets, vegetation, $N_2O$ and $NH_4$, while the atmospheric $pCO_2$ value were adjusted to Younger Dryas levels (Table 2).

### 2.3 Oxygen Isotope Simulation

This study compares simulated $\delta^{18}O$ with correspondent values from marine foraminifera in IODP sediment cores (Dolven et

al., 2002; Hoffmann et al., 2014; Kienast et al., 2001; Kim et al., 2002; Kim et al., 2003; Pelejero et al., 1999; Pivel et al., 2013; Ruehlemann et al., 1999; Steinke et al., 2001) in order to assess whether deviation from the preindustrial climate in response to orbital and $CO_2$ radiative forcing under Younger Dryas conditions are sufficient enough to explain the observed Younger Dryas values. A detailed description of the sedimentary record used in this study is found in Soni (2014) and references therein. Simulated $\delta^{18}O$ of foraminifera ($\delta^{18}O_c$) is estimated from temperature (T) and $\delta^{18}O$ of sea water ($\delta^{18}O_w$),

which is linearly correlated to salinity (S). Salinity is derived from surface (0-250 m) and subsurface water masses (250-1000 m) of the GEOSECS stations in the North Atlantic (Östlund et al. 1987; Labeyrie et al., 1992; Östlund et al., 1987):

$$\delta^{18}O_w = 0.52 * (S - S_0) + \delta_0, \tag{1}$$





$S_0$ =34.9 represents the mean North Atlantic salinity, and $\delta_0$ is set to 0.26 ‰ (Labeyrie et al., 1992, Ostlund et al., 1987). Temperature (T) in seawater is related to $\delta^{18}O$ of foraminifera ($\delta^{18}O_{for.}$; relative to PDB standard) following the equation (O'Neil et al., 1969; Shackleton, 1974; Mix, 1987) including the correction to SMOW following Hut (1987)

$$T = 16.9 - 4.38 * (\delta^{18}O_c - \delta^{18}O_w + 0.27) + 0.1 * (\delta^{18}O_c - \delta^{18}O_w + 0.27)^2, \tag{2}$$

Equation (2) can be rearranged for $\delta_c$ and replacing $\delta_w$ as function of temperature as per equation (2-1), we get the following solution:

$$\delta^{18}O_c = 21.9 \pm \sqrt{(310.61 + 10T)} + \delta^{18}O_w \text{ -0.27}, \tag{3}$$

Equation (3) does allow us to directly compare the simulated $\delta^{18}O_c$ values computed by CESM1 to be compared with the $\delta^{18}O_c$ values from the IODP sedimentary record. There are limitations of this approach due to the assumption of a linear regression between salinity and $\delta^{18}O_w$ which does not consider sea-ice melting and freezing (Mix, 1987). For the Younger

Dryas, additional complications arise regarding $\delta^{18}O_w$ because of the freshwater pulses from the Laurentide Ice sheet, which leads to a non-linear distribution of this tracer in the deep-sea (see e.g. Mikalojewicz et al., 1996). This has been compensated for with a global adjustment of salinity by ~1.0282% in the calculation of simulated $\delta^{18}O_c$.

## 3 Results

### 3.1 Orbital forcing effects on the Younger Dryas climate (YDORB simulation)

#### 3.1.1 Changes in the AMOC strength due to orbital forcing

All parameters used for analysis were averaged annually for the last 50 years of all experiments with a focus on the North Atlantic Ocean region where most of the paleoproxies from marine sediment and ice sheet cores are available. With Younger Dryas orbital parameters, sea surface temperature (SST) values decreased by approximately 1.5 °C for the YDORB simulation compared to the preindustrial simulation, and surface air temperature (SAT) increased over Summit Station, Greenland by

approximately 1.2 °C (Figure 2). The cooling of the SST is attributed to changes in the AMOC, whereas the SAT warming over the region is likely owing to the annual radiation increase. The rate of precipitation over the Northern Atlantic Ocean increased by ~0.3 mm day$^{-1}$ compared to the preindustrial simulation, which resulted in a 0.2 psu decline in sea surface salinity (SSS) relative to the preindustrial simulation (Figure 3). The increased freshwater flux into the North Atlantic Ocean from precipitation, and resulting decline in SSS, leads to a weakening of the AMOC strength of approximately 3.5 Sv compared to

the preindustrial simulation (Figure 4). The idealized age of water masses, which describes the passage of time since a parcel of water has left the surface ocean, supports these results, and shows water mass ages from 250-300 years at the depth of NADW production (Thiele and Sarmiento, 1990; England, 1995; Figure 5).





### 3.1.2 Changes in West African Monsoon Circulation due to orbital forcing

Climate zones discussed in the following for Africa are defined as in Griffiths (1972). Further definition of Region IV is required for the analysis of this study; one north of region V (North Tropics) and one south of region V (South Tropics).

Precipitation for the winter months (December, January, and February) decreased by 3 mm day$^{-1}$ for the equatorial wet (Zone V) region compared to the preindustrial simulation (Figure 6). Winter precipitation rates decreased by approximately 1 mm day$^{-1}$ compared to the preindustrial simulation for the southern tropical region (Figure 6). Precipitation values for the summer (June, July, and August) indicated a 2.5-3 mm day$^{-1}$ increase over the semi-arid region compared to the preindustrial simulation, and precipitation in the northern and southern tropical regions increased by 1 mm day$^{-1}$ and 2.5 mm day$^{-1}$

respectively (Figure 6). There was a 1 mm day$^{-1}$ increase in precipitation over the equatorial wet region (Figure 6).

The location of the intertropical convergence zone (ITCZ), a band of increased precipitation and low-pressure system, alters over the Central African continent (Chiang et al., 2002; Griffiths, 1972) in response to seasonal variations of incoming solar radiation ranging from the Tropic of Cancer during summer to the Tropic of Capricorn during winter (Nicholson, 2013; Waliser

and Gautier, 1993). For the orbital simulation, the high-pressure system is approximately 2 hPa higher than in the preindustrial simulation and is located over the northern tropical region (Figure 7). Over the Mediterranean and northern desert regions (Zones I and II), a low-pressure system covers the region and is approximately 1 hPa lower than in the preindustrial simulation (Figure 7). The convergence of the equatorial northern and southern trade winds, and therefore the ITCZ, is located at approximately 4°S latitude (Figure 7).

The low-pressure system then shifts northward as the solar radiation migrates northward, approaching its northernmost location over the northern desert (Figure 7), and is 4 hPA lower than what is suggested in the preindustrial simulation (Figure 7). The high-pressure system shifted over the southern tropics and is approximately 1012 hPa (Figure 7). The ITCZ from 30°W to 30°E shifted northward to 10 °N latitude (Figure 7).

SAT decreases over the Mediterranean region by ~1 °C during the winter months in the orbital scenario compared to the preindustrial simulation, and 3 °C for the northern desert region (Figure 8), whereas changes in semi-arid region (15°N - 17°N) are more pronounced (-4.5 °C; Figure 8). During the summer months, SAT increases over the northern desert region by approximately 2 °C, and in the Mediterranean region by 5 °C, whereas the semi-arid region is decreased by 3 °C in the orbital

simulation compared to the preindustrial simulation (Figure 8).





### 3.2 Effects of greenhouse gas forcing on the Younger Dryas climate (YDCO2 Simulation)

#### 3.2.1 Changes in the annual AMOC strength due to greenhouse gas forcing

The model indicates a SST anomaly in the North Atlantic Ocean near 30°W and 50°N of -1.2 °C, whereas SAT over Summit Station, Greenland rose by ~2 °C in the YDCO2 scenario compared to the preindustrial control simulation (Figures 2 and 3).

The surface pressure for the area over the Greenland Sea decreased by 0.5 hPa following the change in the temperature gradient. The sea surface salinity anomaly (-0.2 psu) in this region is linked an increase in precipitation (0.2 mm day$^{-1}$) compared to the preindustrial simulation (Figure 3). These differences in buoyancy forcing affect the strength of the AMOC by 2.5 Sv compared to the preindustrial simulation, which is supported by the increase of the ideal age of water mass tracer at the site of NADW production by 10 years (Figures 4 and 5).

#### 3.2.2 Changes in the West African monsoon circulation due to greenhouse gas forcing

Over the northern desert and semi-arid and northern tropical regions, precipitation during the winter months increased by ~0.25 mm day$^{-1}$ compared to the preindustrial simulation, while over the northern southern tropical region, precipitation decreased by ~0.25 mm day$^{-1}$ (Figure 6). With the northward movement of the sun, the precipitation migrates following the ITCZ and the low-pressure systems. Rainfall over the northern tropical region is approximately 0.25 mm day$^{-1}$ less than what is suggested

in the preindustrial simulation, and precipitation also decreases over the equatorial wet and southern tropical regions by ~0.25 mm day$^{-1}$ (Figure 6).

During the winter and summer months, there are insignificant differences in sea surface pressure and location of the ITCZ compared to the preindustrial simulation (Figure 7).

SAT over the winter months is ~0.5 °C warmer compared to the preindustrial simulation and is uniform from the semi-arid to the northern tropical region (Figure 8). The regions to the north and south of the semi-arid and northern tropical regions are ~0.5 °C cooler in the winter months (Figure 8). Similar differences occur for the summer months, though only the western most portions of the semi-arid and northern tropical regions indicate a 0.5 °C warming compared to the preindustrial simulation

(Figure 8). The remainder of the continent shows a ~0.5 °C cooling compared to the preindustrial simulation (Figure 8).

### 4 Discussion

#### 4.1 Atlantic meridional overturning circulation

According to model results, both simulations underestimate the SAT compared to observed values from the GISP2 ice core (Severinghaus et al., 1998). A discrepancy in SST is also depicted in a visual comparison between simulated sea surface

temperatures for both the YDCO2 and YDORB simulations compared to the preindustrial simulation, where sea surface



temperature is consistently 0.5°C – 3.5°C cooler than SST observations from between 11.16 and 13.15 ka (Figure 9; Dolven et al., 2002; Hoffmann et al., 2014; Kienast et al., 2001; Kim et al., 2002; Kim et al., 2003; Pelejero et al., 1999; Pivel et al., 2013; Ruehlemann et al., 1999; Steinke et al., 2001). This is especially true within the waters of Indonesia, as well as the southern Atlantic Ocean off the western coast of Africa and eastern coast of South America (Figure 9). The differences between

the observed data and the preindustrial data indicate that the Younger Dryas was substantially colder than the Holocene, and the simulated SST fails to accurately match the observed data differences. Overall, the comparison indicates that neither scenario is suitable to produce a Younger Dryas like climate. Other climate simulations suggest even further cooling of the SST to approximately 1°C with orbital parameter adjustments (Fischer and Jungclaus, 2010) as well as orbital effects imparting only a small effect on North Atlantic sea surface temperature (Rind et al., 1986). With a cooling of SAT by approximately 15

°C at the onset of the Younger Dryas, the ~1 °C cooling between the Holocene and Younger Dryas by both the YDCO2 and YDORB simulations do not agree with observations inferred by Severinghaus et al. (1998). The model-data bias is likely linked to the neglect of freshwater forcing in our simulation. This study also supports findings a CESM1 simulation for the LGM (Brady et al., 2013). Based on their results, it can be inferred that, for the Younger Dryas, changes in orbital forcing affects temperature more substantially than the greenhouse gas forcing, and the smaller forcing associated with it.

Compared to the preindustrial simulation, precipitation rates over the North Atlantic Ocean increase for the YDORB and YDCO2 by 8.2% and 6.3% respectively, which is not in agreement with the proxy reconstructions that indicate a decrease in storminess during the Younger Dryas cooling event (Alley, 2000). The increase in precipitation for both simulations is likely linked to warmer SAT conditions over the Labrador Sea, and due to mechanisms mentioned earlier, the precipitation increase

is more pronounced in the YDORB simulation.

Results support previous experiments in which orbital sensitivity is linked with increased precipitation over the Northern Atlantic, which is likely a result of warmer SST in the region (Eisenman et al., 2009; Fischer and Jungclaus, 2010). The largest decrease in SSS, compared to the preindustrial simulation, occurs in the YDORB simulation at approximately 35.5 psu, which

is not in agreement with the YD proxy reconstructions of a 27.5-30 psu sea surface salinity (Clark et al., 2001; DeVernal et al., 1996; Ebbesen and Hald, 2004; Schmidt et al., 2004). This decrease in SSS led to a slight decrease of the AMOC in both the YDCO2 and YDORB scenarios, however the YDORB scenario shows a more pronounced weakening (3.5 Sv) compared to the YDCO2 scenario (2.5 Sv). This study is in agreement with previous studies in which changes in orbital parameters only weakly effect AMOC strength (Fischer and Jungclaus, 2010; Gildor and Tziperman, 2000). For example, LGM greenhouse

gas sensitivity experiments indicate changes in greenhouse gases reduce the AMOC strength for the Northern North Atlantic to approximately 23.5 Sv, which is in agreement with the AMOC strength found for the YDCO2 simulation (~24 Sv; Brady et al., 2013). In a comparison between zonally simulated and observed $\delta^{18}O_c$ values, both the YDORB and YDCO2 simulations fail to accurately represent observed values, though the YDORB simulation more closely agrees with the proxy reconstructions (Figure 10). As noted previously, the correlation between observed and simulated $\delta^{18}O_c$ is particularly





primitive, and not likely to be absolutely accurate. This is indicated by the comparison between observed and simulated $\delta^{18}O_c$ for the preindustrial simulation. The values for the observed present day IODP data are more variable compared to the simulated $\delta^{18}O_c$. Because simulated and observed $\delta^{18}O_c$ values have a low correlation for the preindustrial period, correlation between the YDORB and YDCO2 simulations and Younger Dryas minus preindustrial data. However, the trend shown

between the YDCO2 simulation and the YDORB simulation supports findings from previous studies that changes in orbital parameters influences the strength of the overturning circulation more than changes in radiative forcing. It is likely that the stratification of the surface ocean in this study was a result of the freshening of the surface ocean by increased precipitation as well as a pole-ward migration of the sea ice margin due to a warming of the SST. Therefore, AMOC strength is influenced to a greater degree by changes in orbital forcing as opposed to changes in atmospheric $pCO_2$ concentrations; as warming and

precipitation were more pronounced in the YDORB simulation compared to the YDCO2 simulation. However, as shown by the lack of agreement with proxy reconstructions from the Younger Dryas, changes in orbital and atmospheric $pCO_2$ forcing alone are insufficient in producing a climate state consistent with the Younger Dryas. This means that additional forcing changes are required such as freshwater forcing.

**4.2 West African monsoon**

Changes in orbital and atmospheric $pCO_2$ forcing modulate the intensity of the West African monsoon as changes in radiative forcing influence wind, pressure, and temperature, which in turn affects the regional precipitation. The largest change in SAT in the YDORB simulation occurs over the semi-arid and north tropical regions, and the seasonality is more pronounced in the YDORB simulation producing hotter summers and colder winters. The YDCO2 simulation shows little change in SAT

compared to the preindustrial simulation for all regions of the continent, with warming only occurring over the semi-arid and northern tropical regions. Seasonality for the YDCO2 simulation does not change compared to the preindustrial simulation. For the YDORB simulation, maximum temperature in the summer is approximately 39 °C for the Mediterranean and North African desert, and minimum winter temperature is approximately 4 °C for the same regions, which is a 5 °C increase in the summer and 23 °C decrease in the winter. For the YDCO2 simulation, temperature decreased by 0.5 °C in both the winter and

the summer for the same regions compared to the preindustrial simulation.

Gradients in surface pressure systems are modulated by changes in SAT gradients; the stronger the temperature gradient is, the stronger the surface pressure gradient. For the YDORB simulation, the highest-pressure anomaly occurs during the winter over the northern desert region, which coincides with the high SAT anomaly over the same area. Over the same area, surface

pressure increased by ~20 hPa compared to the preindustrial control, which is supported by the high SAT during the summer. For the YDCO2 scenario, the pressure decreases by ~0.5 hPa compared to the preindustrial simulation.





The position of the ITCZ is also influenced by the SAT gradients and resultant pressure anomalies. With increased seasonality in the YDORB simulation, the seasonal position of the ITCZ is also more extreme. For the YDORB simulation, the ITCZ between 30°W and 30°E occurs at approximately 10 °N and 4 °S in the summer and winter respectively. The YDCO2 simulated ITCZ does not appear to migrate compared to the preindustrial simulation. Compared to the preindustrial control, the position

of the ITCZ between 30°W and 30°E for the summer months has migrated ~3° further south in the YDORB simulation. For the winter months, the position of the ITCZ between 30°W and 30°E is the same as the position of the preindustrial simulation for the YDORB. The ITCZ migration is reflected by the convergence of the trade winds and the resultant band of precipitation that develops over the continent. During the summer, the ITCZ between 30°W and 30°E for the YDORB simulation spans the semi-arid to equatorial wet regions (~20 °N to 5 °S). Maximum precipitation occurs at approximately 7 °N and 10°W. For the

winter, the ITCZ centers around 5 °S and spans the equatorial wet to southern tropical regions. The ITCZ between 30°W and 30°E for the YDCO2 simulation is located at 6 °N during the summer and 1 °N during the winter.

Temperature, pressure, and monsoonal precipitation changes are not as pronounced for the YDCO2 simulation, compared to the preindustrial simulation, as the YDORB scenario. The maximum precipitation is approximately 13 mm day$^{-1}$ and 5 mm

day$^{-1}$ for the summer and winter respectively for the YDCO2 simulation, whereas it is approximately 15 mm day$^{-1}$ and 10 mm day$^{-1}$ for the summer and winter respectively in the YDORB simulation.

This study is in general agreement with previous studies (He, 2011; Kitoh et al., 1997; Kutzbach, 1981; Kutzbach et al., 2007; Kutzbach and Guetter, 1986; Otto-Bliesner et al., 2006a; Prell and Kutzbach, 1987; Talbot et al., 2007; Timm et al., 2010;

Trauth et al., 2009; Voss and Mikolajewicz, 2001) that increased insolation in the summer leads to an increase in seasonality and West African monsoonal strength. Additional studies indicate declining greenhouse gases cause precipitation increases in the semi-arid, northern tropical and southern Sahel regions, as well as declines in precipitation due to insolation decreases associated with orbital forcing changes (Otto-Bliesner et al., 2014; Renssen et al., 2006). Cooling in the northern hemisphere could have led to weakening of the African monsoonal system (Alley and Clark, 1999).

It is important to note that the forcing changes in this study do not produce a weakened, dry monsoonal state as seen in YD proxy analysis. This indicates that other forcing mechanisms are required to explain the dry monsoonal state of the Younger Dryas cooling event (Garcin et al., 2007; Gasse, 2000; Lézine et al., 2005; Overpeck et al., 1996; Tierney et al., 2008; Weldeab et al., 2011).

**5 Conclusions**

This study utilized a moderate-resolution comprehensive climate model to determine how the AMOC and West African monsoon respond to Younger Dryas orbital and greenhouse gas forcing. As suggested by the model, the AMOC may be more




affected by Younger Dryas to Holocene changes in seasonal radiative forcing from orbital parameter changes than greenhouse gas changes. This study implies the climate had a higher sensitivity to changes in orbital parameters between the Younger Dryas to Holocene than to relatively low changes in radiative forcing due to atmospheric $pCO_2$ forcing changes. Additionally, orbital and greenhouse gas forcing changes alone were not able to produce the ~15 °C decrease in surface air temperature over

Summit Station, Greenland (Severinghaus et al., 1998) and the weakened AMOC strength needed for a Younger Dryas climate state. This indicates additional forcing must be important to achieve an accurate representation of the Younger Dryas cooling event. The YDORB simulation more closely matches when comparing simulated $\delta^{18}O_c$ with observed $\delta^{18}O_c$ compared to the YDCO2 simulation, indicating that orbital cycles influence the overturning circulation to a higher degree than radiative forcing changes due to atmospheric $pCO_2$. As is seen with the AMOC, the West African monsoon system responds more

profoundly to changes in orbital parameters than changes in atmospheric $pCO_2$ radiative forcing from the Younger Dryas to the Holocene. Surface air temperature, seasonality, wind as well as pressure patterns associated with wind, and the migration of the ITCZ were more pronounced in the YDORB simulation as opposed to the YDCO2 simulation. However, the dry monsoon season suggested by proxy records for the Younger Dryas was not attained through only orbital and greenhouse gas changes. Therefore, other mechanisms are needed for simulations to accurately represent the Younger Dryas climate event.

**Acknowledgements**

We would like to acknowledge the National Science Foundation for their continued support of this project under grants OCE 1536630 and EAR 0903071. This project and these experiments would not be possible without their support. We would also like to thank the National Center for Atmospheric Research for their continued technical support, as well as the continued resources for carrying out the simulations in this project. All simulations for this project were carried out on the National

Center for Atmospheric Research's Wyoming Super Computer Center, supported by the National Science Foundation. We acknowledge the comments from the Editor and reviewers which helped to improve the quality of the manuscript.

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




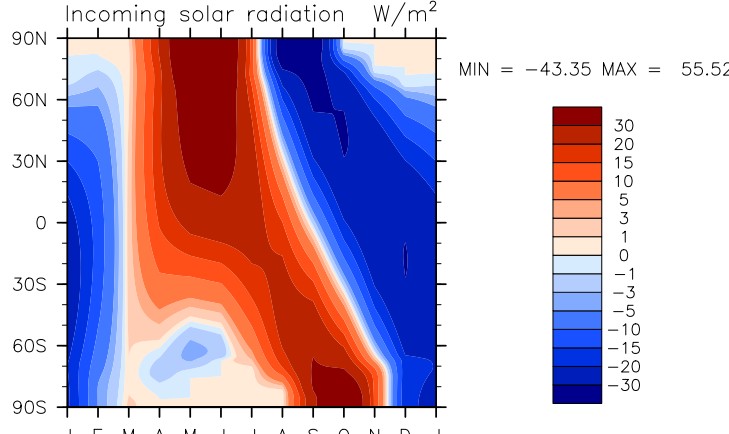

**Figure 1: Radiative forcing differences between the YDORB experiment and the preindustrial simulation in W m⁻² as simulated by CESM1.**





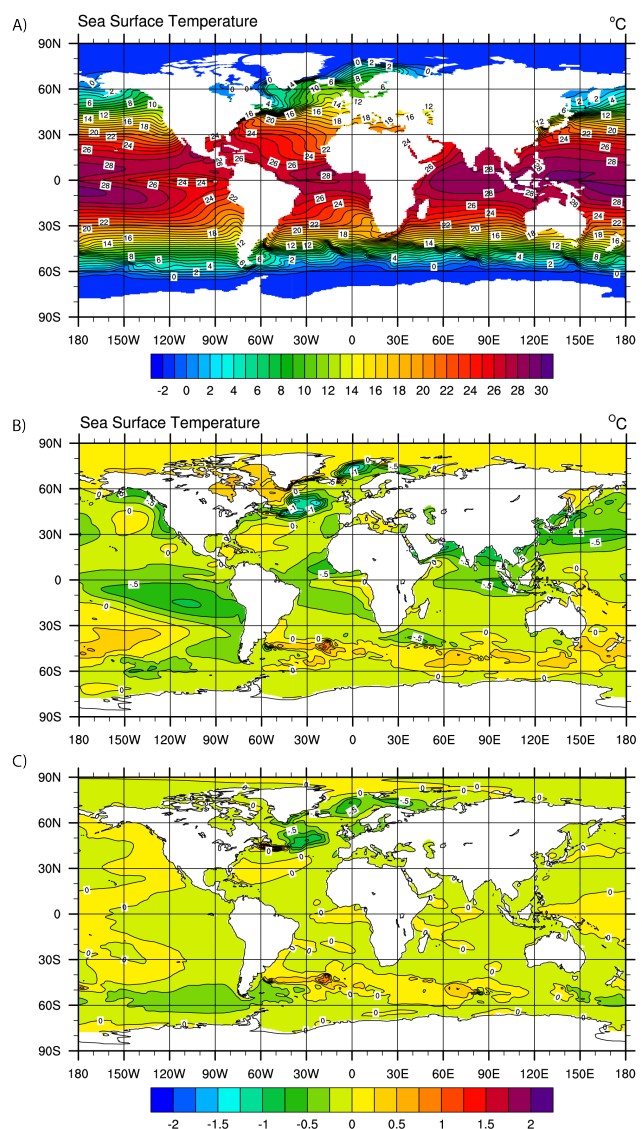

**Figure 2: Sea surface temperature in °C simulated by CESM1 for A) the Preindustrial Control simulation, B) the difference between the YDORB simulation and the Preindustrial Control and C) the difference between the YDCO2 simulation and the Preindustiral Control.**





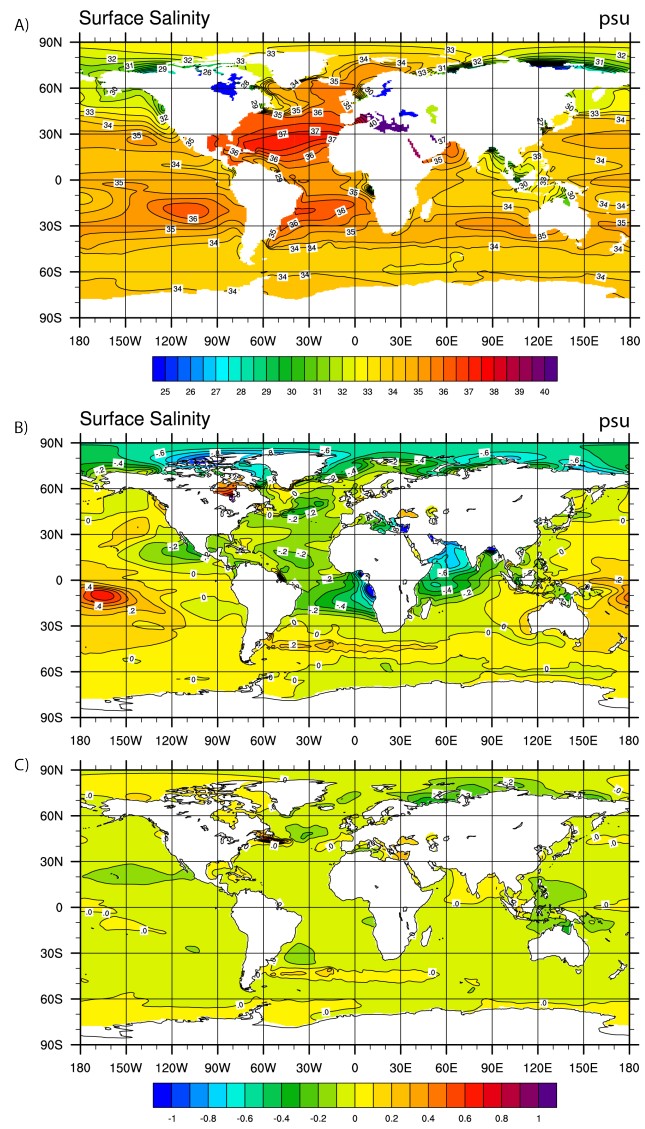

**Figure 3: Sea surface salinity in psu simulated by CESM1 for A) the Preindustrial Control simulation, B) the difference between the YDORB simulation and the Preindustrial Control and C) the difference between the YDCO2 simulation and the Preindustiral Control.**





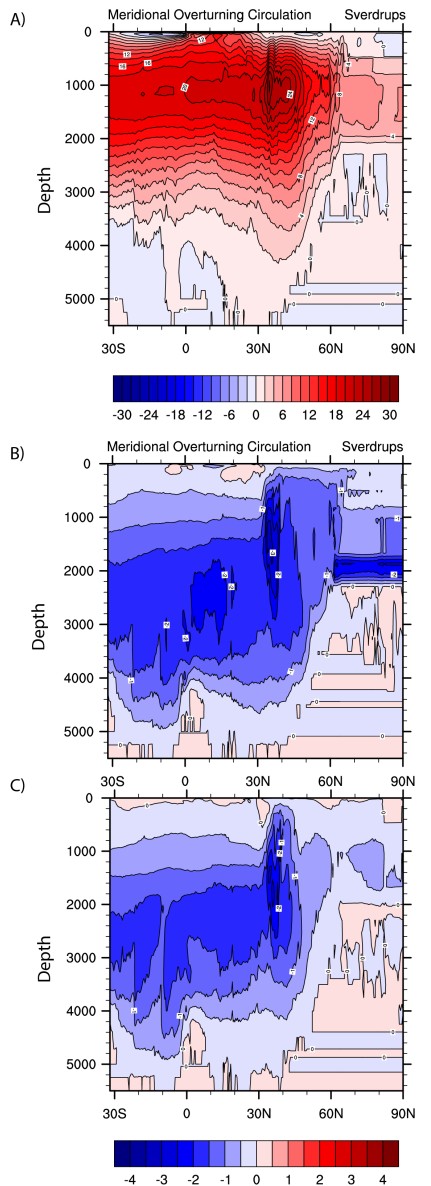

**Figure 4: Meridional overturning circulation in Sv simulated by CESM1 for A) the Preindustrial Control simulation B) the difference between the YDORB simulation and the Preindustrial Control and C) the difference between the YDCO2 simulation and the Preindustrial Control.**





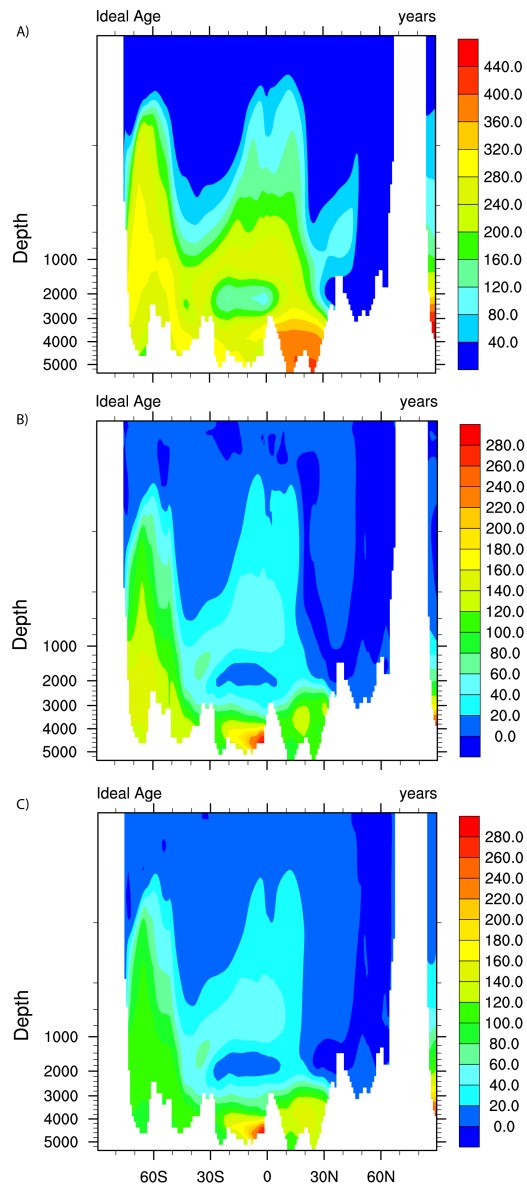

**Figure 5:** Ideal age of water masses in years simulated by CESM1 for A) the Preindustrial Control simulation, B) the difference between the YDORB simulation and the Preindustrial Control and C) the difference between the YDCO2 simulation and the Preindustrial Control.





**Figure 6: Precipitation in mm day$^{-1}$ simulated by CESM1 for A) the Preindustrial Control for the summer months (June, July and August), B) the Preindustrial Control simulation for the winter months (December, January, February), C) the difference between the YDORB simulation and the Preindustrial Control for the summer months, D) the difference between the YDORB simulation and the Preindustrial Control for the winter months, E) the difference between the YDCO2 simulation and the Preindustrial Control for the summer months, and F) the difference between the YDCO2 simulation and the Preindustrial Control for the winter months.**





**Figure 7:** Sea level pressure in hPa and wind velocity vectors in m s$^{-1}$ simulated by CESM1 for A) the Preindustrial Control for the summer months (June, July and August), B) the Preindustrial Control simulation for the winter months (December, January, February), C) the difference between the YDORB simulation and the Preindustrial Control for the summer months, D) the difference between the YDORB simulation and the Preindustrial Control for the winter months, E) the difference between the YDCO2 simulation and the Preindustrial Control for the summer months, and F) the difference between the YDCO2 simulation and the Preindustrial Control for the winter months.





**Figure 8: Surface air temperature in °C simulated by CESM1 for A) the Preindustrial Control for the summer months (June, July and August), B) the Preindustrial Control simulation for the winter months (December, January, February), C) the difference between the YDORB simulation and the Preindustrial Control for the summer months, D) the difference between the YDORB simulation and the Preindustrial Control for the winter months, E) the difference between the YDCO2 simulation and the Preindustrial Control for the summer months, and F) the difference between the YDCO2 simulation and the Preindustrial Control for the winter months.**





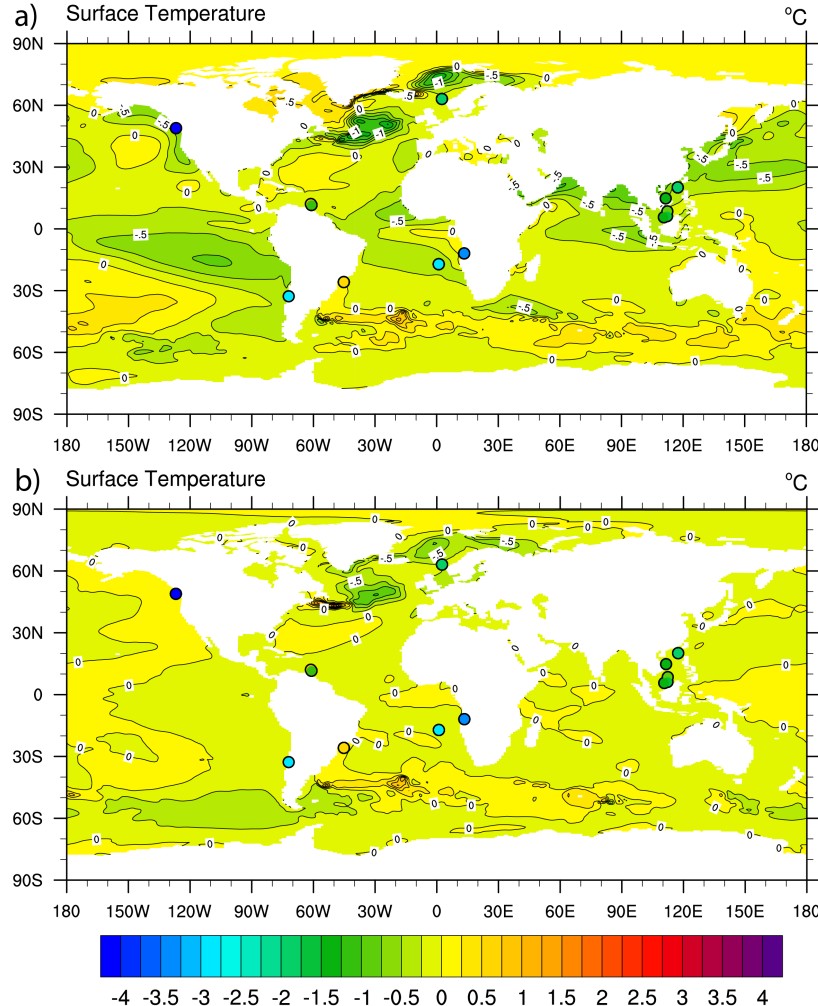

**Figure 9: Difference in sea surface temperature in °C simulated by CESM1 between A) the YDORB simulation and B) the YDCO2 simulation and the preindustrial simulation. For comparison, difference between the Younger Dryas (11.16-13.15 ka) and preindustrial temperature proxies are shown (circles).**



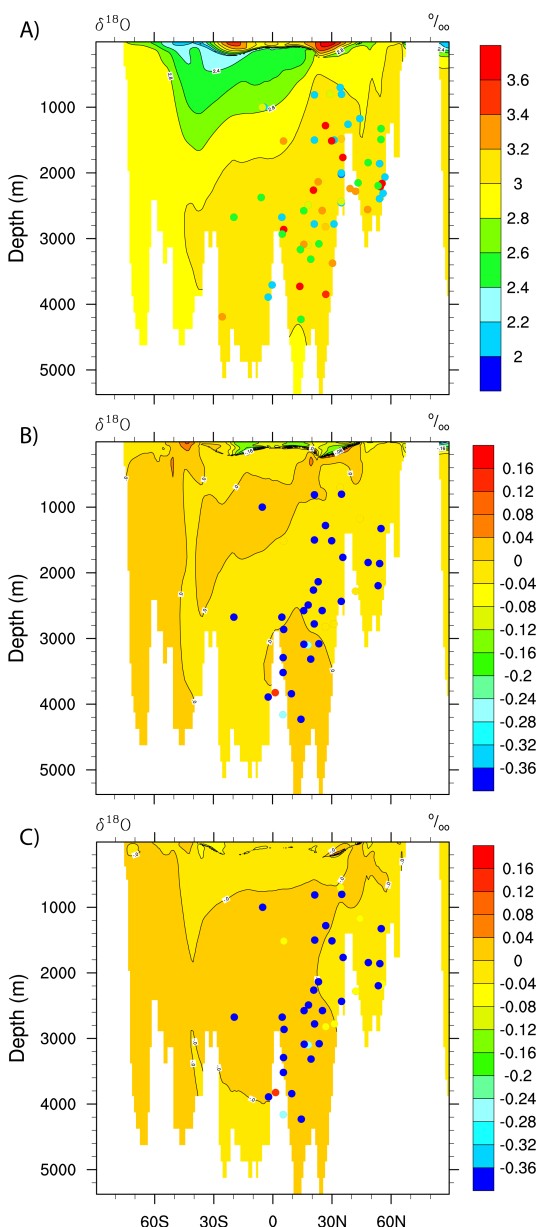

**Figure 10: Distribution of zonal $\delta^{18}O_c$ values in ‰ simulated by CESM1 compared to reconstructed $\delta^{18}O_c$ values from IODP cores (circles; see Soni (2014) and references therein) for the Preindustrial simulation (A). Also shown is the difference between the YDORB and the preindustrial simulation (B), and the difference between the YDCO2 and the preindustrial simulation compared to the difference between the Younger Dryas and preindustrial $\delta^{18}O_c$ data (C).**



| | Incoming Solar Radiation (W m$^{-2}$) | Orbital Parameters | | | Greenhouse Gases | | |
|---|---|---|---|---|---|---|---|
| | | Year | Precession | Eccentricity | Obliquity | CO$_2$ (ppmv) | CH$_4$ (ppbv) | N$_2$O (ppbv) |
| YD | 1,364 | 13 ka | -0.01824 | 0.020175 | 24.093° | 237.6 | 632.0 | 265.0 |
| PI | 1,365 | 1990 | 0.01690 | 0.017236 | 23.446° | 284.7 | 791.6 | 275.7 |

**Table 1. Initial and Boundary Conditions**



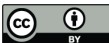

| Simulation | GHGs | Ice Sheet | Orbital Year | Vegetation cover | Aerosol forcing | Length of Simulation |
|---|---|---|---|---|---|---|
| PI | $CO_2$ = 284.7 ppmv<br>$CH_4$ = 791.6 ppbv<br>$N_2O$ = 275.68 ppbv | Modern Greenland and Antarctica | 1990 | Preindustrial | Preindustrial | 100 yr |
| YDCO2 | $CO_2$ = 237.57 ppmv<br>$CH_4$ = 791.6 ppbv<br>$N_2O$ = 275.68 ppbv | PI | PI | PI | PI | 500 yr |
| YDORB | $CO_2$ = 284.7 ppmv<br>$CH_4$ = 791.6 ppbv<br>$N_2O$ = 275.68 ppbv | PI | 13 ka | PI | PI | 500 yr |

**Table 2. Experiment Design**