# Peer review of "The Effects of Younger Dryas Orbital Parameter and Atmospheric pCO2 Changes on Radiative Forcing and African Monsoonal Circulation"

_Climate of the Past, 2018_

## Referee Comment (RC1) · Anonymous Referee #1 · 26 Jun 2018

Overall, the points of the manuscript are not clear. The title said they study the effects of Younger Dryas orbital parameter and atmospheric pCO2 changes on Radiative forcing and African monsoonal circulation. It turns out their focus is AMOC and Younger Dryas climate with some discussion on Africa climate and rarely on radiative forcing. The title needs to be changed.

Page 1 line 25, it seems that neither orbital parameter nor pCO2 is the dominant cause of Younger Dryas climate. They are at second or higher order. This needs to be clarified. Page 5. Line 5 where is Table 4-1? Page 5. Line 10. The experiment design is quite vague. My guess is that the initial condition of these experiments is taken from a

1000-year simulation performed by NCAR. While the authors change the forcing fields of orbital forcing and GHGs. There are one control simulation and two sensitivity simulations in total. In table 2, I see several different descriptions, Preindustrial, PI, Modern Greenland and Antarctica,..., please clarify these items. Overall the experiment design parts need to be rewritten.

Page 5. 2.3 Oxygen Isotope Simulation. This part may be not needed.

Page 6. Add and compare the numbers from previously published freshwater experiments. Show AMOC time series of these experiments.

Overall, the results section is too descriptive and lack of physical explanation.

Page 9. Line 28, then what's new for this study?

The abstract also needs to be rewritten to clarify the points.

---

## Referee Comment (RC2) · Anonymous Referee #2 · 29 Jun 2018

**Summary Statement**

The article is presenting a model-based analysis of the response of the climate system to external forcing from two of the major factors that affected the Earth's climate during the last deglaciation process and the transition period into the Holocence. In this time domain falls the millennial-scale 'event' of the Younger Dryas associated with a major reorganization in the Atlantic Ocean circulation (the Atlantic meridional overturning circulation, AMOC) and significant cooling in the Northern Hemisphere. Although a number of different causes have been proposed as explanations for the YD (e.g. Schenk et al., doi:10.1038/s41467-018-04071-5 ref. therein) the reduction in AMOC

strength is still a working hypothesis for explaining the YD climate anomaly. The climatic impacts of a reduction in AMOC and the internal feedback mechanisms have been studied with models of intermediate complexity and both low and high-resolution climate models such as CESM1 before.

Here the authors present another modeling study with CESM concentrating on the African monsoonal circulation. It is noteworthy that the authors do not focus on the meltwater forcing in their modeling experiments. They instead choose to study the individual impacts of atmospheric greenhouse gas concentrations ($CO_2$) and orbital forcing. They compare their modeling results with proxy data and one of their main conclusion is that neither orbital nor $CO_2$ forcing alone cannot produce simulated anomalies in the monsoonal circulation large enough to be consistent with the proxy data. The proxy-model comparison is - contrary to the title and the main results presented in the article - using a global network of marine temperature proxies and proxies related with the oxygen isotope signal of the AMOC.

The authors use the model simulations to investigate the changes in the monsoonal circulation over Africa using temperatures and rainfall and dynamical variables to describe the atmospheric circulation response.

**General review**

My overall impression was that the article is lacking a well-stated working hypothesis. It is not clear to me if the authors wanted to test the AMOC-shutdown as cause for the YD event (via meltwater input into the North Atlantic) or whether they wanted to demonstrate that orbital forcing and GHG forcing are sufficient to generate a YD event. In one case the idea would be to reject the hypothesis, in the latter case they wanted to find modeling support for the GHG/orbital forcing hypothesis. The way the conclusion is written in the abstract (last two sentences), that is not quite clear.

Irrespective of what was the ultimate objective in their research, the simulations can only provide a partial analysis regarding the causes. I don't think one can gain significant new insight into the forcing and feedbacks and regional monsoonal response without adding at least a freshwater forcing simulation to the list of experiments. The authors should consider if a factor separation method can be applied in a more formal way with additional forcing experiments.

In conclusion, I find the current version needs a major revision at least, but I would rather suggest a re-submission of a manuscript that starts with a working hypothesis and presents the additional sensitivity experiments, or else a strong supportive statement must be made why the two simulations can provide new insight into YD climate.

**Detailed comments**

**Introduction**

page 2 (paragraph 2, 3 ), page 3 (paragraph 1,2):

The authors begin the introduction with the review of the potential causes for the YD, particularly the freshwater forcing. Then they explain the role of orbital forcing for the direct control of the monsoonal circulation, followed by the GHG section. In those paragraphs the authors could highlight more clearly the relation with the direct impact on the monsoonal circulation and distinguish from that the impact on the AMOC with the associated (indirect) impact on the monsoonal circulation.

**Methods**

**Model description and experiments**

Comment for this section:

Additional information would be important to mention for the study of the YD: Does the CESM simulation include interactive vegetation as part of the land model?

Specific comments:

p.4 line 29-32:

Please mention if the climate sensitivity to CO2 forcing is model-specific or a more general estimate. If so, mention uncertainty associated with the temperature response (and radiative forcing).

p.5 l.1-3: Please mention that the insolation anomalies are from the CESM model using the present-day calendar convention (that is a change in the length of seasons is not accounted for).

p.5 section 2.2: I suggest you mention here already the length of the simulations and the years chosen for averaging.

p.5 section 2.3 (minor issue): I was not sure is the model simulation calculating the $\delta^{18}O$ *online* or are those equations applied to the generated output temperature and salinity data afterwards.

**Results**

p.6 30-32: On the age of water masses:

How did you calculate the age? Was the 'clock' for interior ocean waters reset to zero at the start of each simulation, or did you keep the age from the restart ocean state? I wonder because Fig.5 shows that the oldest water masses are not older than 500 years. So, the 500-year simulations are arguably too short to get the steady-state of the deep ocean and the age of the water masses.

p.7 l. 21: "solar radiation migrates northward" : explain more carefully which features of the incoming radiation migrates northward. Is it a specific summer integrated value, a local maximum, or ...?

p.7. l. 23-24 This part could be combined with the northward migration of the ITCZ and the description of the pressure fields in lines 15-20.

p.8 section 3.2.2:

Are you referring to anomalies relative to the PI mean state in the migration of the

seasonal insolation? If yes, please refer to them as anomalies. Else explain how the rainfall anomalies are related to the mean PI seasonal insolation in the CO2 forcing experiment.

**Discussion**

I miss a discussion of internal feedbacks here. E.g. how important are changes in ocean SSTs or land surface characteristics (vegetation feedbacks) in general.

p.9 line 10-15:

The section should be written with a working hypothesis in the center of the discussion. Following my earlier comments on the lack of a clear working hypothesis, it is not particularly informative to discuss the orbital and GHG forcing experiments as a 'evidence' against the freshwater hypothesis. The reason I mention this is that we have to be careful with how to draw logical conclusions here: We pick a forcing that results in a mismatch between model simulation and proxies. That is evidence against the hypothesis that the applied forcing was the cause for the changes observed in proxies during YD. But alternative explanations (such as the freshwater forcing) are not 'supported' by this negative outcome (we could likewise cite here the less prominent hypothesis of an impact event as a cause for YD, too)

That is also the reason why I would recommend to conduct the additional freshwater experiments (and the factor separation method).

p.9 l.10-15: I would exercise caution with comparing LGM with YD climate states and response to forcing. However, more important if you want to make this point then explain briefly why the LGM is a good time period to study the response to orbital forcing and to compare it here to the YD time.

p.9-p.10 last line to first line: What is 'primitive' referring to in the correlation results?

p.10. l. 24: The winter temperature change is very large in response to orbital forcing. Worth to be discussed (realistic, do we have a chance to find evidence for the

temperature changes in regional proxies?)

p.11, l. 12-16: Anomalies with respect to PI control simulation would be useful to add, too.

**Conclusion**

The conclusion is lacking a strong message. The authors mention again that the two individual forcing experiments do not simulate the YD climate state consistent with proxies from ocean and land. However, they cannot say much more because they miss the freshwater forcing case (and other combinations of the individual forcing factors). It is not particularly useful to then continue the conclusion with a comparison which of the two simulations is closer to the proxies. Certainly one could say more positively that orbital forcing causes large changes in the monsoonal circulation directly, therefore freshwater forcing plays a lesser role for the North African monsoonal climate.

**Reference list**

P. 16. l. 10: check author: "F. and Lu, Z."

**Figures**

In many figures the contour interval labels are using a font that is rather small and hard to read. (in particular Fig.4 )

Figure 5: Water masses: caption should explain the cross section? What longitudes / ocean basin?

Figure 7: Something is wrong in this figure panel: The wind vectors and SLP pattern do not align. I believe you mixed the SLP and wind pattern (e.g. figure C shows same winds a figure B and results in unphysical non-geostrophic flow components) Same problem in Figures D,E,F.

And caption: What atmospheric level is the wind field?

Figure 9: This figure should be accompanied by scatter plots using nearest model grid-box and the corresponding proxy values.